# Links between meaning in life and physical quality of life after rehabilitation: Mediating effects of positive experiences with physical exercises and mobility

Katarzyna Czekierda[1☯‡]*, Karolina Zarychta[1☯‡], Nina Knoll[2], Jan Keller[2], Aleksandra Luszczynska[1,3]*

**1** Wroclaw Faculty of Psychology, SWPS University of Social Sciences and Humanities, Wroclaw, Poland, **2** Department of Psychology, Freie Universität Berlin, Berlin, Germany, **3** Trauma, Health, & Hazards Center, University of Colorado at Colorado Springs, Colorado Springs, CO, United States of America

☯ These authors contributed equally to this work.
‡ These authors share first authorship on this work.
* kczekierda@swps.edu.pl (KC); aluszczy@uccs.edu (AL)

**Data Availability Statement:** The datasets used and/or analyzed during the current study are available at https://osf.io/q3yar/.

## Abstract

### Background

Indicators of emotional processes (positive experiences with physical exercises) and functional processes (mobility) were previously found to be associated with positive cognitive resources (meaning in life), and the key outcome in the rehabilitation, namely physical quality of life (QOL). Yet, the mediating roles of such processes were not tested. Therefore, this prospective study investigated whether the relationship between meaning in life and physical QOL was mediated by positive experiences with physical exercises and mobility.

### Methods

Prospective data were collected at two measurement points, 1 month apart. A total of $N = $ 339 participants (aged 19–84 years old, 57.9% women) provided data at Time 1 (T1) at the beginning of inpatient rehabilitation from central nervous system diseases (CNSD, e.g., stroke; $n = 89$) or musculoskeletal system diseases (MSD, e.g., dorsopathies; $n = 250$), and $n = 234$ at Time 2 (T2, the end of rehabilitation; 4 weeks after T1). Mediation analysis with meaning in life as predictor (T1), positive experiences with physical exercises and mobility as sequential mediators (T2), and physical QOL (T2) as the outcome was conducted.

### Results

Higher meaning in life (T1) predicted more positive experiences with physical exercises (T2), which were associated with a higher level of mobility (T2), which in turn was associated with better physical quality of life (T2).

**Funding:** The study was funded by the National Science Centre, Poland, grant no. 2014/15/B/HS6/00923, awarded to AL. The funders had no role in study design, data collection and analysis, decision to publish, or preparation of the manuscript.

**Competing interests:** The authors have declared that no competing interests exist.

## Conclusions

Meaning in life at the beginning of inpatient rehabilitation may trigger positive experiences with physical exercises and functional changes in mobility levels, leading to better physical quality of life. Screening for low meaning in life may allow to identify patients who are at risk for a lack of improvement of mobility and physical quality of life during rehabilitation.

## Introduction

Central nervous system diseases (CNSD) and musculoskeletal system diseases (MSD) are among the most prevalent chronic conditions [1]. They result in physical functioning limitations regarding mobility, dexterity, behavioral problems, daily living activities, and psychosocial functioning [1, 2]. A common consequence of CNSD and MSD is a decline of physical quality of life (QOL) [3–5]. The World Health Organization's International Classification of Functioning, Disability, and Health (WHO ICF) [6] presents disability, functioning, and health as the outcomes of interactions between health conditions and contextual factors (i.e., environmental and personal factors). Physical QOL reflects people's perception of their physical state [7] and is conceptualized as an indicator of physical health [6].

To improve their functional and physical health, people living with CNSD and MSD usually require extensive rehabilitation (e.g., physical exercises) [8, 9]. This is supported by a large body of evidence showing that physical exercises, including rehabilitation exercises, improve various health indicators among people with CNSD [10, 11] and MSD [12, 13]. Although CNSD and MSD are two different groups of chronic conditions, both have in common that persons living with CNSD and MSD face mobility problems [14], show a decline in physical QOL [3, 4], and benefit from rehabilitation exercises regarding their mobility and physical QOL [14].

Meaning in life, defined as the purposeful and meaningful engagement with life, is considered to be a determinant of psychological functioning [7] and eudaimonic well-being [15, 16]. Models of well-being [17] assume that meaning in life may be associated with positive affective reactions to people's positive experiences [17, 18]. Moreover, meaning in life was found to predict physical QOL among people with chronic conditions [19, 20]. According to the stroke recovery cycle [21], meaning in life is associated with physical QOL through better positive emotional processes (e.g., positive experiences with physical exercises), and better daily functioning (e.g., higher mobility) [21]. Specifically, the model suggests that stroke severity, patients' emotional state, and sense of meaning in life are the key predictors of physical QOL after stroke [21]. Additionally, the model [21] implies that post-stroke rehabilitation interventions should focus on promoting post-stroke functioning (e.g., improving patients' mobility), restoring and promoting meaning in life, and improving physical QOL.

Participation in rehabilitation exercises may result in positive experiences with exercises or satisfaction with experienced behavior and its outcomes, which in turn may improve behavioral maintenance [22, 23, 24]. Among patients with chronic low back pain who were prescribed a six-weeks individualized physical exercise program, 89% reported positive experiences with physical exercises [25]. Qualitative research conducted in the context of chronic illness showed associations between participation in rehabilitation exercise programs, positive experiences with physical exercises, and physical QOL [26]. Positive experiences with physical exercises during cardiac and orthopedic rehabilitation predicted long-term physical exercise maintenance after rehabilitation [24]. Physical activity programs which prompt

positive experiences with physical exercises are also likely to result in a reduction of functional limitations in mobility levels in older adults [27].

Although direct relationships between positive experiences with physical exercises and physical QOL are possible, the associations between these two constructs could be indirect, mediated by mobility. Mobility is defined as the ability to get around and is one of the facets of physical QOL reflecting the scope of independence [6]. Mobility is a significant correlate of physical QOL among people with disabilities [28]. According to the model of joyful movement [27], positive experiences with physical exercises are related to increased mobility. The model was confirmed in a longitudinal study enrolling older adults [27].

To summarize, the association between meaning in life and physical health indicators such as physical QOL are well-documented [19, 20, 29]. However, the remaining question is whether positive experiences with physical exercises as well as functional changes in mobility levels would sequentially mediate the relation between meaning in life and physical QOL in the context of inpatient rehabilitation. In line with the stroke recovery cycle [21] it was hypothesized that higher levels of meaning in life (the independent variable, measured at the beginning of the inpatient rehabilitation) would be associated with higher of levels positive experiences with physical exercise (the first mediator). Furthermore, in line with the model of joyful movement [27] it was hypothesized that the positive experiences with physical exercise (the first mediator), would be related to better physical QOL (the dependent variable, measured at the end of the inpatient rehabilitation), via mobility (the second mediator). In sum, it was expected that meaning in life would be associated with higher physical QOL at the follow-up, through sequential mediators, positive experiences with physical exercises (the first mediator) and higher levels of mobility (the second mediator).

## Materials and methods

In the present study, the prospective association between meaning in life (measured at the beginning of the inpatient rehabilitation) and physical QOL (measured at the end of the inpatient rehabilitation, the 1-month follow-up) is investigated in the context of CNSD or MSD. Two sequential mediators (positive experiences with physical exercises and mobility) were assumed to operate between meaning in life and physical QOL.

### Participants

At Time 1 (T1), $N$ = 339 participants (57.9% female) aged 19–84 years old ($M$ = 54.41, $SD$ = 11.32; 85% aged 19–65 years old, with 57% of patients with CNSD aged $\leq$ 65 years old, and 96% of patients with MSD aged $\leq$ 65 years old) responded to the baseline questionnaire assessing variables under study. At Time 2 (T2, 1 month later), $n$ = 234 participants (59.8% female) aged 19–84 ($M$ = 54.83, $SD$ = 12.03) responded to a second questionnaire. The total attrition rate was 31%. Descriptive characteristics of the sample (including the type of disease) are provided in Table 1. Further details referring to the subtypes of CNSD and MSD are presented in S1 Table.

### Procedure

The study was approved by the Internal Review Board at the first author's institution, SWPS University of Social Sciences and Humanities, Wroclaw, Poland. Written informed consents were obtained from all participants. Participants were recruited among patients who were admitted to 6 inpatient physical or neurological rehabilitation wards in South-West Poland. All patients at the respective rehabilitation centers who had neurological or musculoskeletal system illnesses were invited to participate. The inclusion criteria were: (1) being at least 18

**Table 1. Descriptive characteristics of the sample.**

| | Time 1 (total *N* = 339) | Time 2 (*n* = 234) |
|---|---|---|
| | % (*n*) | % (*n*) |
| People with central nervous system diseases (*n*, %) | 26.3% (89) | 35.5% (83) |
| People with musculoskeletal diseases (*n*, %) | 73.7% (250) | 64.5% (151) |
| Marital status | | |
| married or in a long-term relationship | 71.4% (242) | 69.8% (163) |
| Single | 28.6% (97) | 31.2% (73) |
| Education level | | |
| primary or vocational education | 36.1% (122) | 32.9% (77) |
| secondary education | 41.7% (142) | 42.3% (99) |
| higher education degree | 22.2% (75) | 24.8% (58) |
| Employment status | | |
| full-time or part-time employment | 59.5% (202) | 50.0% (117) |
| unemployed/pensioner/retired | 40.5% (137) | 50.0% (117) |
| Economic situation (compared with the average family in the country) | | |
| similar | 56.6% (192) | 57.2% (134) |
| better | 23.9% (81) | 24.4% (57) |
| worse | 19.5% (66) | 18.4% (43) |
| Place of residency | | |
| urban area | 85.8% (291) | 63.7% (149) |
| rural area | 14.2% (48) | 36.3% (85) |

years old, and (2) for participants with CNSD: a lack of severe cognitive impairment (i.e., scores > 10) measured with the Montreal Cognitive Assessment (MoCA [30]). On average, participants with CNSD scored *M* = 22.67, *SD* = 4.81 (range 10–30) on the MoCA scale.

Inpatient rehabilitation was funded by the public nation-wide insurance system (i.e., participants were not charged) and lasted at least 21 days. After an initial period of 21 to 28 days, patients' condition was evaluated and followed by an extension of rehabilitation for up to six months. The CNSD rehabilitation program aimed at improving kinesthetic/movement abilities and condition, counteracting spasticity, improving self-care, self-management, cognitive function, speech improvement, and health behavior change education. The MSD rehabilitation program aimed at improving kinesthetic/movement abilities and condition, movability of joints, physical strength, physical flexibility, pain management, and health behavior change education.

Time 1 (T1) took place at the beginning of the rehabilitation. Time 2 (T2) data collection was conducted 1 month later, at the end of patients' rehabilitation. Patients were informed about the study aims, design, and anonymity. All potential respondents, who were invited and met the inclusion criteria, agreed to participate. Data collection was conducted individually and lasted approximately 30 minutes, respectively for T1 and T2. In case they had problems with responding to questionnaires (e.g., because of problems with reading), participants were interviewed by the study personnel.

## Measures

*Meaning in life* was assessed at T1 with one item: 'To what extent do you feel your life to be meaningful?' derived from the World Health Organization Quality of Life measure [6].

Participants were asked to provide their answers on a 5-point scale ranging from 1 ('not at all') to 5 ('extremely'). Using 1-item measures of meaning in life yielded similar results as using more complex measures [29].

*Physical QOL* was assessed at T1 and T2 with 6 items derived from the WHOQOL-BREF questionnaire (physical QOL domain [6]). Participants were asked to provide their answers to questions (e.g., 'To what extent do you feel that physical pain prevents you from doing what you need to do?') on a 5-point scale ranging from 1 ('not at all') to 5 ('extremely'). One item from the physical QOL domain of the WHOQOL-BREF: 'How well are you able to get around?' represented mobility and was therefore not included in the present physical QOL measure. The reliability of the measure was low at T1, with $\alpha$ = .45, and acceptable at T2, with $\alpha$ = .70. Higher scores represent better physical quality of life.

*Positive experiences with physical exercises* were measured at T1 and T2 with 1 item from the Exercise Experiences Subscale of Health Related Experiences [23], 'When I was physically active, I experienced that I felt better afterward'. Participants were instructed to refer to the experiences with exercises performed during the inpatient rehabilitation. Responses were given on a 4-point scale ranging from 1 ('definitely no') to 4 ('definitely yes').

*Mobility* was measured at T1 and T2 with 1 item: 'How well are you able to get around?'. This item was derived from the World Health Organization Quality of Life measure [6]. Responses were given on a 5-point scale ranging from 1 ('not at all') to 5 ('extremely'). The 1-item measurement of mobility was used in previous research [31].

Socio-demographic variables were measured at T1. Participants were asked to indicate their education level on a 3-point scale ranging from 1 ('primary education') to 3 ('at least 5 years of higher education/MA or MSc'). Similarly, they indicated their perceived economic status on a 3-point scale ranging from 1 ('my economic situation is worse than the average economic situation of a family in the country') to 3 ('my economic situation is better than the average economic situation of a family in the country'). Furthermore, participants indicated their employment status by checking 'yes' (when having a full-time or a part-time employment) or 'no' (when being unemployed/retired/pensioner). Lastly, they were asked about how many months have passed since they received their CNSD or MSD diagnosis.

## Data analysis

The G*Power calculator [32] was used to determine the sample size. Assuming medium effect sizes ($f^2$ = 0.15) and accounting for potential confounders, the sample size was estimated to include at least 300 participants. Data were analyzed using IBM SPSS, version 25 [33].

To test whether the relationship between patients' meaning in life (T1) and physical QOL (T2) was sequentially mediated by positive experiences with physical exercises (T2; the first sequential mediator) and mobility (T2; the second sequential mediator), we performed multiple mediation analyses with sequential mediators using PROCESS with 10,000 bootstraps. Model 6 was applied. This model allows for testing indirect effects of the independent variable on the dependent variable, assuming that the independent variable predicts the first mediator, which predicts the second mediator. In turn, the second mediator predicts the dependent variable (the outcome) [34]. PROCESS allows the testing of mediator hypotheses by assuming that mediators are operating together in a sequence (i.e., positive experiences with physical exercises would predict mobility). In line with MacKinnon [35] it was assumed that the following significant associations are essential to establish a mediation: (1) between the independent variable and the first mediator, (2) between the first mediator and the second mediator, and (3) between the second mediator and the dependent variable. The significant association between the independent and dependent variables is not an essential condition, because the expected

mediation effects are of medium size [35]. The analyses were conducted controlling for base-line levels of the T2 variables; controlling for the baseline levels of the mediator and dependent variables is the recommended approach, in particular if there are less measurement points than three [35].

Little's MCAR test indicated that the missing data patterns were systematic, Little's $\chi^2(357)$ = 475.69, $p < .001$. To reduce the potential negative impact of systematic dropout, missing data were accounted for with a maximum likelihood estimation procedure, recommended for data with systematic attrition [36]. The total sample ($N$ = 339) was analyzed and all missing data were accounted for by using full information maximum likelihood procedures.

## Results

### Attrition analysis

Completers did not differ significantly from those who dropped out regarding meaning in life, physical QOL, age, gender, time since diagnosis, perceived economic status and education level. Completers and those who dropped out differed in terms of employment status, with completers reporting being employed more often (57.6%) than dropouts (42.4%). They also differed in T1 mobility levels, with completers reporting lower levels of mobility ($M$ = 3.36, $SD$ = 0.88) than drop-outs ($M$ = 3.56, $SD$ = 0.66), and in terms of the type of diagnosis (patients with MSD dropped out more often [39.6%] than those with CNSD [6.7%]). For details of attrition analyses see S2 Table.

### Differences between CNSD and MSD patients, descriptive analysis, and correlations between study variables

As displayed by S3 Table, patients with MSD (vs patients with CNSD) reported higher levels of meaning in life at T1 and mobility (T1 and T2). There were no between-group differences in physical QOL levels at T1. At T2, patients with MSD reported higher physical QOL levels compared to patients with CNSD. There were no between-group differences in positive experiences with physical exercises at T1 or T2.

Descriptive statistics and correlations among study variables are reported in Table 2. The hypothesized outcome, mediators, and the independent variable formed significant positive bivariate associations.

### Testing the hypothesized mediation model

The results obtained for the multiple mediation model (Fig 1 and Table 3) showed that the association between meaning in life (T1) and physical QOL (T2) was sequentially mediated by

**Table 2. Descriptive statistics and correlations between the study variables at T1 and T2.**

| | | $M$ ($SD$) | 2 | 3 | 4 | 5 | 6 | 7 |
|---|---|---|---|---|---|---|---|---|
| 1 | Meaning in life (T1) | 3.77 (0.91) | .25*** | .40*** | .24*** | .29*** | .33*** | .34*** |
| 2 | Physical quality of life (T1) | 2.94 (0.51) | | .46*** | .29*** | .66*** | .36*** | .30*** |
| 3 | Mobility (T1) | 3.42 (0.82) | | | .38*** | .45*** | .53*** | .36*** |
| 4 | Positive experiences with physical exercises (T1) | 3.06 (0.71) | | | | .29*** | .33*** | .56*** |
| 5 | Physical quality of life (T2) | 3.12 (0.55) | | | | | .58*** | .43*** |
| 6 | Mobility (T2) | 3.47 (0.71) | | | | | | .42*** |
| 7 | Positive experiences with physical exercises (T2) | 3.09 (0.61) | | | | | | |

\*\*\*$p < .001$. Abbreviations: T1: Time 1 (beginning of inpatient rehabilitation); T2: Time 2 (the 1-month follow-up, at the end of inpatient rehabilitation); $M$: mean; $SD$: standard deviation.

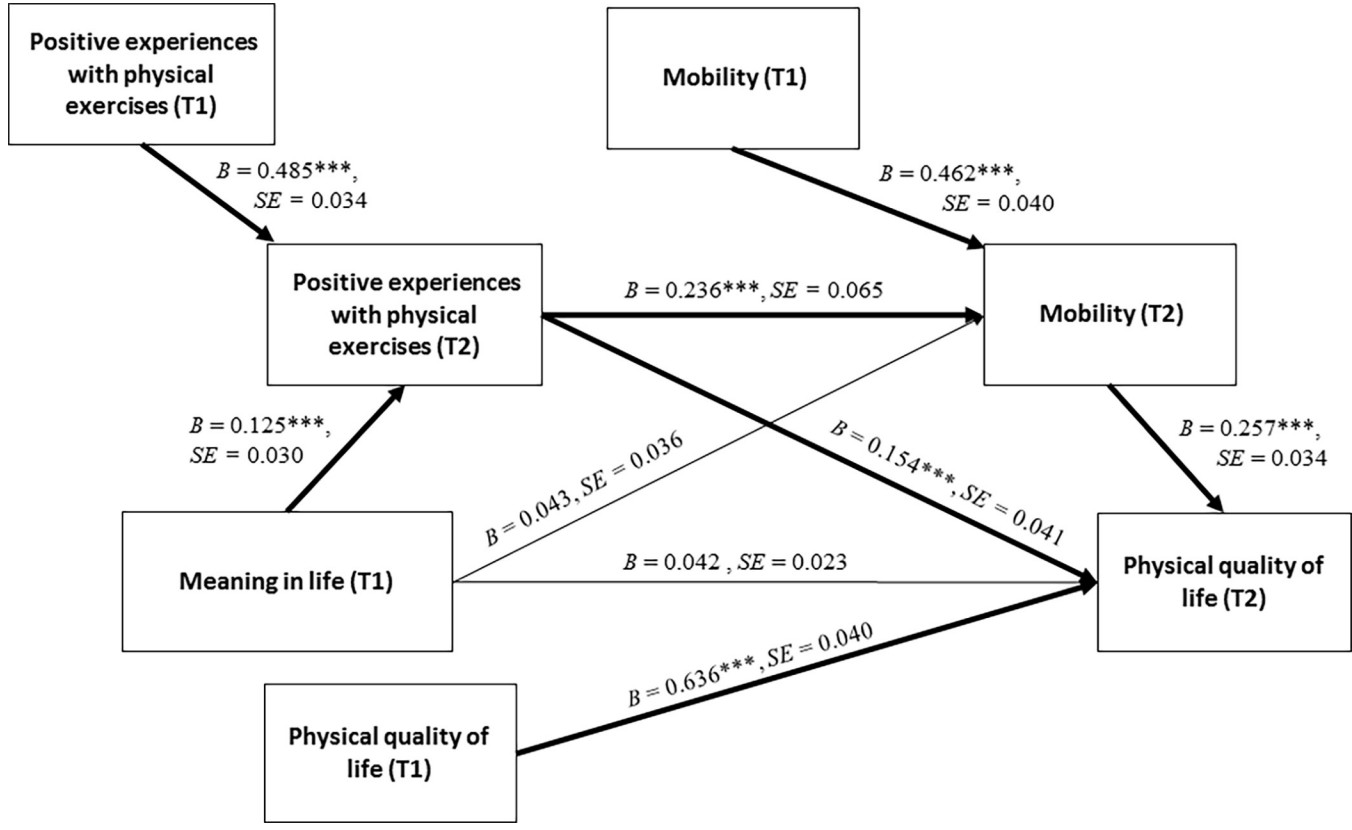

**Fig 1. The hypothesized sequential mediation model explaining physical QOL among 339 people with CNSD or MSD.** ***$p < .001$. Abbreviations: T1: Time 1, the baseline (at the beginning of inpatient rehabilitation); T2: Time 2, the 1-month follow-up (at the end of inpatient rehabilitation).

positive experiences with physical exercises (T2; the first sequential mediator) and mobility (T2; the second sequential mediator), as indicated by significant indirect effects (Table 3). In particular, participants who reported higher levels of meaning in life at the beginning of the inpatient rehabilitation were likely to report more positive experiences with physical exercises (T2), which were related to a higher level of mobility (T2), which, in turn, was associated with better physical QOL at the end of the inpatient rehabilitation. Moreover, a model with positive experiences with physical exercises (T2) acting as a single mediator between meaning in life (T1) and physical QOL (T2) was also found to be significant.

The hypothesized model was tested for a second time, controlling for potential confounders: patients' age and gender, time since the diagnosis, and the type of diagnosis (CNSD vs MSD). A similar pattern of findings emerged, with a significant indirect effect for the two sequential mediators (positive experiences with physical exercises as the first mediator, mobility as the second mediator), $B = 0.008$, $SE = 0.004$, 95% CI [0.002, 0.017].

## Discussion

This prospective study provides novel evidence for the indirect association between patients' meaning in life and their physical QOL in the context of inpatient rehabilitation among people with MSD and CNSD. In particular, patients with CNSD or MSD who reported higher levels of meaning in life were more likely to perceive more positive experiences with physical exercises (T2; the first sequential mediator) and higher levels of mobility (T2; the second sequential mediator), and consequently reported better physical QOL (T2).

**Table 3. Effects of meaning in life on physical quality of life through sequential mediators.**

| Indirect effects pathways | *B* | *SE* | 95% CI | |
|---|---|---|---|---|
| | | | Lower | Higher |
| **Meaning in life (T1) → Positive experiences with physical exercises (T2) → Physical QOL (T2)** | **0.019** | **0.007** | **0.006** | **0.036** |
| Meaning in life (T1) → Mobility (T2) → Physical QOL (T2) | 0.011 | 0.009 | -0.007 | 0.031 |
| **Meaning in life (T1) → Positive experiences with physical exercises (T2) → Mobility (T2) → Physical QOL (T2)** | **0.007** | **0.003** | **0.002** | **0.016** |
| Direct effect pathways | | | | |
| **Meaning in life (T1) → Positive experiences with physical exercises (T2)** | **0.125** | **0.030** | **0.066** | **0.184** |
| Meaning in life (T1) → Mobility (T2) | 0.043 | 0.036 | -0.029 | 0.115 |
| Meaning in life (T1) → Physical QOL (T2) | 0.042 | 0.023 | -0.002 | 0.088 |
| **Positive experiences with physical exercises (T2) → Mobility (T2)** | **0.236** | **0.065** | **0.107** | **0.364** |
| **Positive experiences with physical exercises (T2) → Physical QOL (T2)** | **0.154** | **0.041** | **0.073** | **0.236** |
| **Mobility (T2) → Physical QOL (T2)** | **0.257** | **0.034** | **0.190** | **0.324** |
| **Positive experiences with physical exercises (T1) → Positive experiences with physical exercises (T2)** | **0.485** | **0.034** | **0.409** | **0.562** |
| **Mobility (T1) → Mobility (T2)** | **0.462** | **0.040** | **0.384** | **0.540** |
| **Physical QOL (T1) → Physical QOL (T2)** | **0.636** | **0.040** | **0.556** | **0.716** |

Values of indirect effect coefficient (*B*) presented in bold are significant. Confidence intervals (CI) were calculated with the bootstrapping method, based on 10,000 repetitions. CI that do not include zero indicate a significant indirect effect. Significant coefficients are marked in bold. Abbreviations: QOL: quality of life; T1: Time 1 (the beginning of inpatient rehabilitation); T2: Time 2 (the 1-month follow-up, at the end of inpatient rehabilitation).

The results of the present study are in accord with previous research indicating that positive experiences with physical exercises are related to mobility [27], or that mobility is related to physical QOL [37]. However, previous research allowed for relatively limited conclusions as the majority used cross-sectional designs and focused on bivariate associations. In contrast, the present study provides preliminary evidence for a chain of associations and indirect effects of positive experiences with physical exercises and mobility, that operate linking meaning in life and a key outcome of rehabilitation, physical QOL.

Our study provides support for Ryff's model of well-being [17] that assumes that meaning in life may be closely related to the positive affective reaction of people's positive experiences [17, 18]. The assumptions of the stroke recovery cycle [21] were also confirmed. We found preliminary evidence that meaning in life may be associated with better physical QOL indirectly, through more positive experiences with physical exercises.

The findings of the present study are also in line with the meaning model proposed by Park et al. [19]. This model highlights the role of meaning in life as the central cognitive resource determining emotional and behavioral adaptation processes, and leading to better physical QOL among patients with chronic conditions [19]. Some potential implications for clinical practice may be drawn. For example, screening the levels of meaning in life at the beginning of rehabilitation may allow to identify people who are subsequently at risk for less positive experiences with physical exercises, lower mobility, and in turn, lower physical QOL. Experimental research showed that a meaning making intervention delivered to patients with a chronic illness may improve patients' meaning in life for up to three months after an intervention [38]. However, such interventions did not have a direct effect on a QOL indicator, accounting for the physical domain [38]. These findings [38] may be interpreted as partially in line with our findings, suggesting that the associations between meaning in life and physical QOL are

indirect (mediated by other constructs) rather than direct. The results of the present study suggest that the potential mechanism, explaining the link between meaning in life and physical QOL, may include positive experiences with physical exercises and improved mobility. This assumption, however, should be tested in further experimental research.

The present study has several limitations. The time gap between the measurement points was short, so it was not possible to test for long-term effects. The results should be treated as preliminary evidence, due to a suboptimal number of measurement points for a sequential mediation model. Accounting for 3 or 4 measurement points would allow drawing more in-depth conclusions and such a design is recommended for the future studies. Another limitation refers to attrition rates that were high, albeit similar to attrition rates observed in other research, conducted in the context of inpatient and outpatient rehabilitation [23]. The patterns of missing data were systematic. To reduce the potential negative impact of high and systematic dropout, missing data were accounted for with a maximum likelihood estimation procedure, recommended for data with systematic attrition [36] and dropout rates as high as 50% [39].

A further limitation refers to the 1-item measurement of meaning in life, mobility, and positive experiences of physical exercises. These constructs should be assessed using measures with more items, however, a meta-analysis indicated that previous research which used one-item measures of meaning in life yielded similar results to those which used a more complex assessment [29]. Single item measurement of mobility was successfully applied in previous research [31, 32]. Regarding the 1-item measure of positive experiences with physical exercises, exploratory factor analyses showed that this particular item loads at the subscale of health-related positive experiences with a factor loading value of .78 [23]. Internal consistency of the measure of physical QOL at T1 was low. This was due to scores of 1 item (#16, sleep quality), that had a distinctly skewed distribution ($g_1$ = 0.67) with 58% of participants indicating that the quality of their sleep was low or very low whereas other items had a normal distribution. Low scores reduced the reliability of the scale ($\alpha$ = .70 after the deletion of item #16). However, the item was retained for the purpose of future comparisons across studies using the standard 7-item version of the WHOQOL-BREF scale. The present study did not account for other factors which were found to be associated with physical QOL (e.g., social support or self-efficacy [40]). Future research may test the mediation model controlling for these factors. The subsample of participants with CNSD was relatively small, which did not allow for a well-powered exploratory analysis, testing if the type of diagnosis matters. Furthermore, the analyzed sample was ethnically homogeneous (all participants were white), and thus generalizations to ethnically diverse populations should be made with caution. Cognitive decline was used as an exclusion criterion among patients with CNSD, but it was not applied among patients with MSD. Cognitive decline is more likely to co-occur with MSD and functional/mobility issues, but mostly in people aged > 65 years old [41]. Only 4% of people with MSD in our sample were > 65 years old. Still, cognitive decline could occur among people with MSD participating in the present study, and consequently, affect the findings. Finally, it is possible that social desirability has contributed to potential biases in participants' responses.

## Conclusions

This prospective study confirmed the role of positive experiences with physical exercises and mobility as factors sequentially mediating the association between meaning in life and physical QOL among patients in rehabilitation. Higher levels of meaning in life measured at the beginning of inpatient rehabilitation were associated with better physical QOL at the end of the 1-month rehabilitation, through higher levels of positive experiences with physical exercises

and higher levels of mobility. Screening for meaning in life at the beginning of inpatient rehabilitation may allow to identify individuals who are more likely to improve physical QOL.

## Supporting information

**S1 Table. Sample characteristics for subgroups based on ICD-10 diagnosis.** Abbreviations: ICD 10: The International Statistical Classification of Diseases and Related Health Problems, 10th revision (WHO, 2016), *M*: mean, *SD*: standard deviation.
(DOCX)

**S2 Table. Results of attrition analysis.** All variables were measured at Time 1. Abbreviations: *M*: mean; *SD*: standard deviation; [a] *p*-values in bold indicate statistical significance; [b] economic status was measured by a 3-point scale: 1—the economic situation worse than the average family in Poland, 2—similar to the economic situation of an average family in the country, 3 –better; [c] education was measured by a 3-point scale: 1—primary education or vocational education (no high school education), 2—secondary education, 3—higher education; [d] employment was measured by a 2-point scale: 1—employed (full- or part-time), 2—unemployed (or being retired or a pensioner); [b, c, d] for percentage of categories for economic status, education level, and employment see Table 1 (the manuscript).
(DOCX)

**S3 Table. Differences in study variables between subsamples with central nervous system diseases (CNSD) and musculoskeletal system diseases (MSD).** [a] *p*-values in bold indicate statistical significance. Abbreviations: QOL: Quality of life; T1: Time 1 (the beginning of inpatient rehabilitation); T2: Time 2 (1-month follow-up, at the end of inpatient rehabilitation); CNSD: central nervous system disease; MSD: musculoskeletal system disease; *M*: mean; *SD*: standard deviation; *df*: degrees of freedom; $\eta^2$: partial eta squared.
(DOCX)

## Author Contributions

**Conceptualization:** Katarzyna Czekierda, Aleksandra Luszczynska.

**Data curation:** Nina Knoll.

**Formal analysis:** Katarzyna Czekierda, Karolina Zarychta, Aleksandra Luszczynska.

**Funding acquisition:** Aleksandra Luszczynska.

**Investigation:** Katarzyna Czekierda.

**Methodology:** Katarzyna Czekierda, Karolina Zarychta, Nina Knoll, Jan Keller, Aleksandra Luszczynska.

**Project administration:** Aleksandra Luszczynska.

**Supervision:** Nina Knoll, Aleksandra Luszczynska.

**Writing – original draft:** Katarzyna Czekierda, Karolina Zarychta, Nina Knoll, Jan Keller, Aleksandra Luszczynska.

**Writing – review & editing:** Katarzyna Czekierda, Karolina Zarychta, Jan Keller, Aleksandra Luszczynska.

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
