## [Decision Letter · Decision Letter 0]

21 Aug 2019

PONE-D-19-21036

Links between meaning in life and physical quality of life after rehabilitation: Mediating effects of positive experiences with physical exercises and mobility

PLOS ONE

Dear Mrs. Czekierda,

Thank you for submitting your manuscript to PLOS ONE. After careful consideration, we feel that it has merit but does not fully meet PLOS ONE’s publication criteria as it currently stands. Therefore, we invite you to submit a revised version of the manuscript that addresses the points raised during the review process.

We would appreciate receiving your revised manuscript by Oct 05 2019 11:59PM. To enhance the reproducibility of your results, we recommend that if applicable you deposit your laboratory protocols in protocols.io, where a protocol can be assigned its own identifier (DOI) such that it can be cited independently in the future. For instructions see: http://journals.plos.org/plosone/s/submission-guidelines#loc-laboratory-protocols

We look forward to receiving your revised manuscript.

Kind regards,

Stefan Hoefer

Academic Editor

PLOS ONE

Journal Requirements:

1. Meaning in life and physical quality of life: Cross-lagged associations during inpatient rehabilitation.

https://psycnet.apa.org/doiLanding?doi=10.1037%2Frep0000281

In your revision ensure you cite all your sources (including your own works), and quote or rephrase any duplicated text outside the methods section. Further consideration is dependent on these concerns being addressed.

Additional Editor Comments (if provided):

Thank you very much for submitting your manuscript. We received three reviews and based on their recommendations and my own reading of the manuscript it may be considered for publication after fully addressing all of the points raised by each of the three reviewers.

Reviewers' comments:

Reviewer's Responses to Questions

**Comments to the Author**

1. Is the manuscript technically sound, and do the data support the conclusions?

Reviewer #1: Yes

Reviewer #2: Yes

Reviewer #3: Partly

2. Has the statistical analysis been performed appropriately and rigorously? 

Reviewer #1: Yes

Reviewer #2: Yes

Reviewer #3: I Don't Know

3. Have the authors made all data underlying the findings in their manuscript fully available?

Reviewer #1: Yes

Reviewer #2: Yes

Reviewer #3: Yes

4. Is the manuscript presented in an intelligible fashion and written in standard English?

Reviewer #1: Yes

Reviewer #2: Yes

Reviewer #3: Yes

5. Review Comments to the Author

Reviewer #1: The paper on links between meaning in life and physical quality of life after rehabilitation has a number of strengths, in particular the study's longitudinal research design and the large sample of rehabilitation patients. Assessment tools are suboptimal as most of them consist of single item measures. Also, the reliability of one 6-item measure was low. Authors are aware of these two shortcomings, and they mention this as a limitation.

When analyzing mediating effects of positive experiences with physical exercises and mobility, three of the four variables in the chain were assessed cross-sectionally. Mediation with cross-sectional data is adversarial to the spirit of mediation modeling, and violates implicit model assumptions. Authors are aware of this, and they mention this as a limitation.

At Time 2, 234 patients had returned. One might want to mention this in the Abstract? The total sample (N = 339) was analyzed and all available data were accounted for using full information maximum likelihood procedures. Information about the status of missings could be added. Missing values at random?

Table 2 (Descriptive statistics and correlations between the study variables at T1 and T2) has a mistake. The numbering in horizontal line is wrong: category 5 is missing.

In Line 215: here the CI needs to be repaired: , 95%BCI = [.002; .017], see APA style. At least the B needs to be removed.

Most of the shortcomíngs mentioned in this review cannot be targeted in a revision. However, the paper is overall interesting and well done, making a contribution to the literature. The minor technical flaws should be corrected.

Reviewer #2: The manuscript presents a longitudinal study that examined sequential mediation between meaning in life to positive affect in PA, then to mobility and then to QoL. There are 2 measurement point, less than the ideal for sequential analysis, yet an improvement over most cross sectional studies.

The paper is on an important issue of rehabilitation among patients with chronic conditions, is well written and analyzed in a sophisticated and rigorous fashion. The discussion acknowledges all the weaknesses of the paper (e.g., 2 time points, short measures etc).

Minor.

Table 1 should have also the N in each variable and not only the percentage.

Methods:

Reliability of α=0.45 is not acceptable (p. 8) as the text implies. There is an explanation in the discussion, but this does not make the value acceptable.

References # 30 and #31 are work that may also have used the 1-item mobility measure but are probably not the origin of the measure. Pls explain.

Recommendation: accept with minor revisions.

Reviewer #3: This study investigated the association between the constructs of ‘meaning of life’ and ‘physical quality of life’ in inpatients receiving rehabilitation; and explored if this association was mediated by mobility and having positive experiences of exercise. The authors applied mediation methods described by Hayes on a sample of 339 participants at baseline soon after admission to rehabilitation and n=239 four weeks later. The results were interpreted to suggest that the positive links between meaning of life and physical QOL were mediated by positive experiences of exercise and functional changes in physical quality of life.

The results should be reviewed by an expert in mediation analysis but from my simple understanding, the fact that direct pathways between meaning of life and mobility, and between meaning of life and physical QOL were nonsignificant (Table 3), suggests that mediation should not proceed further. That, is if these direct pathways are not significant (not different form zero) does it makes sense to ask if they are mediated by another factor?

Another issue that may require further explanation is the rationale for the links between the factors investigated, and hence the possible clinical implications (accepting the mediation results reported). For example, what is the logical or hypothesised causal link between the construct of meaning of life and functional changes in mobility? Also, it the construct of meaning of life amenable to change or does it just allow inpatients in rehabilitation to be screened?

Some specific comments

Abstract, line 20: Perhaps replace ‘general resource’ with ‘construct’.

Introduction, line 32: Please define the construct of ‘physical quality of life’. How does this construct map onto the WHO ICF?

Introduction, line 56: Not sure of the intended meaning of this sentence. Is it suggesting that making the experience of exercise positive is an important outcome of rehabilitation? Many would argue the purpose of exercise is to lead to functional improvement, that enjoyment, as such, is not an outcome.

Methods, line 98: Very high drop out rate which is a limitation of the study (31%).

Methods, line 110: Why was cognitive decline not an exclusion criterion for patients with MSD?

Methods, line 145: When the participant answered the item about positive experiences, what was their time frame. For example, could they be referring to the past before their health episode that led to rehabilitation?

Data analysis, lines 165- 170: More detail required. Was a sample size estimation completed? How were assumptions with modelling tested? A brief explanation of model 6 would help the reader

6. PLOS authors have the option to publish the peer review history of their article (what does this mean?). If published, this will include your full peer review and any attached files.

Reviewer #1: No

Reviewer #2: Yes: Efrat Neter

Reviewer #3: No

---

## [Author Response · Author response to Decision Letter 0]

26 Sep 2019

Reviewers' comments to the Authors:

Reviewer #1: 

1. The paper on links between meaning in life and physical quality of life after rehabilitation has a number of strengths, in particular the study's longitudinal research design and the large sample of rehabilitation patients. Assessment tools are suboptimal as most of them consist of single item measures. Also, the reliability of one 6-item measure was low. Authors are aware of these two shortcomings, and they mention this as a limitation.

Authors’ response: Thank you for the positive evaluation of the design of the study and careful reading. We agree with the limitations of the study, yet we hope (as the reviewer indicated) they were appropriately discussed in the limitations section. Furthermore, measuring these constructs with single items was often applied in the past and resulted in findings similar to those that were obtained with more complex measures (for a meta-analysis see Czekierda et al., 2018, Health Psych Rev). 

2. When analyzing mediating effects of positive experiences with physical exercises and mobility, three of the four variables in the chain were assessed cross-sectionally. Mediation with cross-sectional data is adversarial to the spirit of mediation modeling, and violates implicit model assumptions. Authors are aware of this, and they mention this as a limitation.

Authors’ response: Thank you for this comment. The revised manuscript (the data analysis section) indicates that the approach with two measurement points and controlling for the baseline levels of mediators and the dependent variable is acceptable, although suboptimal (see MacKinnon, 2008).

3. At Time 2, 234 patients had returned. One might want to mention this in the Abstract? The total sample (N = 339) was analyzed and all available data were accounted for using full information maximum likelihood procedures. Information about the status of missings could be added. Missing values at random?

Authors’ response: Thank you. The information about the sample size at T2 is now provided in the abstract. The revised Data Analysis section clarifies that Little’s MCAR test indicated that the missing data patterns were systematic, Little’s χ2(357) =,475.69 p < 001. To reduce the potential negative impact of systematic dropout, missing data were accounted for with a maximum likelihood estimation procedure, recommended for data with systematic attrition [36]. The total sample (N = 339) was analyzed and all available data were accounted for using full information maximum likelihood procedures.

4. Table 2 (Descriptive statistics and correlations between the study variables at T1 and T2) has a mistake. The numbering in horizontal line is wrong: category 5 is missing.

Authors’ response: Thank you for careful reading; the numbering has been corrected.

5. In Line 215: here the CI needs to be repaired: , 95%BCI = [.002; .017], see APA style. At least the B needs to be removed.

Authors’ response: Thank you, CI reporting has been corrected in line with APA style and reported as “95% CI [0.002, 0.017]”.

6. Most of the shortcomíngs mentioned in this review cannot be targeted in a revision. However, the paper is overall interesting and well done, making a contribution to the literature. The minor technical flaws should be corrected.

Authors’ response: We very much appreciate the recognition of the strengths of the studies and all the comments made by the Reviewer.

Reviewer #2: 

1. The manuscript presents a longitudinal study that examined sequential mediation between meaning in life to positive affect in PA, then to mobility and then to QoL. There are 2 measurement point, less than the ideal for sequential analysis, yet an improvement over most cross sectional studies. The paper is on an important issue of rehabilitation among patients with chronic conditions, is well written and analyzed in a sophisticated and rigorous fashion. The discussion acknowledges all the weaknesses of the paper (e.g., 2 time points, short measures etc.).

Authors’ response: Thank you for the positive evaluation of the study design and possible impact of the findings. 

Minor.

2. Table 1 should have also the N in each variable and not only the percentage.

Authors’ response: Thank you, respective numbers have been added.

Methods:

3. Reliability of α=0.45 is not acceptable (p. 8) as the text implies. There is an explanation in the discussion, but this does not make the value acceptable.

Authors’ response: The Reviewer is right. The interpretation of the size of the coefficient was changed as follows: “The reliability of the measure was low at T1, with α = .45 and acceptable at T2, with α = .70”

4. References # 30 and #31 are work that may also have used the 1-item mobility measure but are probably not the origin of the measure. Pls explain.

Authors’ response: Thank you for careful reading. The respective paragraph was revised. We indicate thatMobility was measured at T1 and T2 with1 item: ‘How well are you able to get around?’ This item was derived from the World Health Organization Quality of Life measure [6]. Responses were given on a 5-point scale ranging from 1 (‘not at all’) to 5 (‘extremely’). The one-item measurement of mobility was used in previous research [31].

5. Recommendation: accept with minor revisions.

Authors’ response: Thank you for your helpful suggestions.

Reviewer #3: 

1. This study investigated the association between the constructs of ‘meaning of life’ and ‘physical quality of life’ in inpatients receiving rehabilitation; and explored if this association was mediated by mobility and having positive experiences of exercise. The authors applied mediation methods described by Hayes on a sample of 339 participants at baseline soon after admission to rehabilitation and n=239 four weeks later. The results were interpreted to suggest that the positive links between meaning of life and physical QOL were mediated by positive experiences of exercise and functional changes in physical quality of life.

Authors’ response: Thank you for thoroughly reading the manuscript. We appreciate all insightful comments and suggestions made by the Reviewer.

2. The results should be reviewed by an expert in mediation analysis but from my simple understanding, the fact that direct pathways between meaning of life and mobility, and between meaning of life and physical QOL were nonsignificant (Table 3), suggests that mediation should not proceed further. That, is if these direct pathways are not significant (not different form zero) does it makes sense to ask if they are mediated by another factor?

Authors’ response: Thank you for this comment. As the Reviewer noticed, Baron and Kenny (1986; earlier: Judd & Kenny, 1981; James & Brett, 1984) recommended that establishing a significant association between the independent and dependent variable is essential to establish a mediation. Respectively, in the sequential mediation, a significant association between the independent variable and the second mediator would be essential to establish a sequential mediation, via the first mediator.

However, Kenny, Kashy and Bolger (1998), and subsequently other authors (see e.g., Frazier, Tix, & Barron, 2204; MacKinnon, 2008; MacKinnon, Fairchild & Fritz, 2008) suggested that the direct association between the independent and dependent variable should be dropped. Among others, the assumption strongly reduces the likelihood of detecting the mediation which is moderate or small in size (MacKinnon et al., 2008) and therefore the direct association between the independent and dependent variable may be considered only if the assumed mediating effect is of large size.

Based on previous research, there was no background to expect large effect sizes, therefore we did not assume significant direct effects between the independent and the dependent variable (or, respectively, between the independent variable and the second mediator).

These issues were clarified in the revised Data Analysis section. We indicate thatIn line with MacKinnon [35] it was assumed that the following significant associations are essential to establish a mediation: (1) between the independent variable and the first mediator, (2) between the first mediator and the second mediator, and (3) between the second mediator and the dependent variable. The significant association between the independent and dependent variables is not an essential condition, because the expected mediation effects are of medium size [35].

Frazier, PA., Tix., AP., Barron, KE. (2004). Testing moderator and mediator effects in counseling psychology research. Journal of Counseling Psychology, 51, 115-134.

MacKinnon DP. Introduction to statistical mediation analysis. New York: Lawrence Erlbaum Associates; 2008.

MacKinnon, D.P., Fairchild, A J., & Fritz, MS (2008). Mediation Analysis. Annual Review of Psychology, 58, 593-514.

3a. Another issue that may require further explanation is the rationale for the links between the factors investigated, and hence the possible clinical implications (accepting the mediation results reported). For example, what is the logical or hypothesised causal link between the construct of meaning of life and functional changes in mobility? Also, it the construct of meaning of life amenable to change or does it just allow inpatients in rehabilitation to be screened?

Authors’ response: Thank you for raising these relevant questions. In our study, we did not hypothesize direct associations between meaning in life and mobility. In contrast, we assumed that the association between positive experiences with physical exercise is linked to mobility, which, in turn is associated with better physical QOL. 

To clarify this issue, the study aims section was revised to indicate thatIn line with the stroke recovery cycle [21] it was hypothesized that higher meaning in life (the independent variable) measured at the beginning of inpatient rehabilitation would be associated with higher reports of positive experiences with physical exercise (the first mediator). Furthermore, in line with the model of joyful movement [27] it was hypothesized that the positive experiences with physical exercise (the first mediator), would be related to better physical QOL (the dependent variable), via mobility (the second mediator). In sum, it was expected that meaning in life would be associated with higher physical QOL at the follow-up, through two sequential mediators, positive experiences with physical exercises (Mediator 1) and higher levels of mobility (Mediator 2).

3b. Also, it the construct of meaning of life amenable to change or does it just allow inpatients in rehabilitation to be screened?

Authors’ response: Thank you for this interesting question. The discussion section was revised in order to address interventions that target meaning in life. 

In particular, we indicate that for example, screening the levels of meaning in life at the beginning of rehabilitation may allow to identify people who are subsequently at risk for less positive experiences with physical exercises, lower mobility, and in turn, lower physical QOL. Experimental research showed that meaning making interventions delivered to patients with a chronic illness are effectively improving meaning in life for up to three months after an intervention [38]. However, such intervention did not have direct effect on QOL indicator, accounting for the physical domain [38]. These findings [38] may be interpreted as partially in line with our findings, suggesting that the associations between meaning in life and physical QOL are indirect, mediated by other constructs. Consequently, a psychosocial intervention may boost meaning in life among patients with chronic illnesses and low levels of meaning in life. Our results of the present study suggest that the potential mechanism explaining the link between boosted meaning in life and physical QOL may include positive experience with physical exercises and improved mobility. This assumption, however, should be tested in further experimental research.

4. Some specific comments

Abstract, line 20: Perhaps replace ‘general resource’ with ‘construct’.

Authors’ response: Thank you, respective sentence was revised and shortened.

5. Introduction, line 32: Please define the construct of ‘physical quality of life’. How does this construct map onto the WHO ICF?

Authors’ response: Thank you. Physical QOL is now defined in the context of WHO ICF in the Introduction as follows: “The World Health Organization’s (WHO) International Classification of Functioning, Disability, and Health (ICF) [6] presents disability, functioning, and health as the outcomes of interactions between health conditions and contextual factors (i.e., environmental and personal factors). Physical QOL reflects people’s perception of their physical state [7] and it is conceptualized as an indicator of physical health, and one of personal factors in the ICF [6].”

6. Introduction, line 56: Not sure of the intended meaning of this sentence. Is it suggesting that making the experience of exercise positive is an important outcome of rehabilitation? Many would argue the purpose of exercise is to lead to functional improvement, that enjoyment, as such, is not an outcome.

Authors’ response: Thank you for this comment. We agree with the Reviewer that the respective sentence was unclear and could be interpreted in various ways. The revised version includes a corrected sentence, which is in line with research and models of behavioral maintenance (e.g. Rothman, 2010; Fleig et al. 2011).

 In particular, we state that participation in rehabilitation exercises may result in positive experience with exercises or satisfaction with experienced behavior and it outcomes, which in turn improves behavioral maintenance [22, 23, 24].

7. Methods, line 98: Very high dropout rate which is a limitation of the study (31%).

Authors’ response: Thank you for this relevant comment. The Discussion section was revised to clarify respective limitations, such as, the large scale of dropout. We also clarify that the best available methods to combat high dropout with MAR patterns were applied. 

We have also indicated that although high, the attrition is similar to previous research conducted in the in-patient and out-patient rehabilitation. For example, Fleig et al. (2011) enrolled 415 patients at the beginning of rehabilitation with 78% participating 2 weeks later and 60% participating at approximately 8 weeks after the baseline.

The revised Discussion section clarifies that another limitation refers attrition rates which were high, albeit similar to attrition observed in other research conducted in the context of in-patient and out-patient rehabilitation [23]. Furthermore, the patterns of missing data were systematic. To reduce the potential negative impact of high and systematic dropout, missing data were accounted for with a maximum likelihood estimation procedure, recommended for data with systematic attrition [36] and dropout as high as 50% [39].

The revised Data Analysis section clarifies that Little’s MCAR test indicated that the missing data patterns were systematic, Little’s χ2(357) = 475.69, p < .001. To reduce the potential negative impact of systematic dropout, missing data were accounted for with a maximum likelihood estimation procedure, recommended for data with systematic attrition [36]. 

8. Methods, line 110: Why was cognitive decline not an exclusion criterion for patients with MSD?

Authors’ response: Among participants with MSD, 95% were ≤ 65 years old.

The age of patients with CNSD and MSD was clarified in the revised Methods section: “At Time 1 (T1), N = 339 participants (57.9% female) aged 19 – 84 (M = 54.41, SD = 11.32; 85% aged 19-65 years old, with 57% of CNSD patients aged ≤ 65 years old, and 98 MSD patients aged ≤ 65 years old).”

Cognitive decline is more likely to co-occur with MSD and functional/mobility issues in people aged > 65 (Calero-Garcia et al., 2016). Therefore, we did not consider that this may be a factor causing a bias in the present study.

However, we agree with the Reviewer that this is yet another limitation of the study. Consequently, this issue was addressed in the revised Discussion section. We state that cognitive decline was used as an exclusion criterion only for patients with CNSD, and not applied for patients with MSD. Cognitive decline is more likely to co-occur with MSD and functional/mobility issues in people aged > 65 [41], and 96% of people with MSD in our sample were ≤ 65 years old. However, it cannot be excluded that cognitive decline could occur among people with MSD participating in the present study, and consequently, affect the findings.

9. Methods, line 145: When the participant answered the item about positive experiences, what was their time frame. For example, could they be referring to the past before their health episode that led to rehabilitation?

Authors’ response: Thank you for this question. Participants were instructed to refer to their experience with exercise during the rehabilitation. The revised Methods section includes the respective information. 

10. Data analysis, lines 165- 170: More detail required. Was a sample size estimation completed? How were assumptions with modelling tested? A brief explanation of model 6 would help the reader.

Authors’ response: Thank you for these questions. Sample size was estimated by G*Power calculator, which is now stated in the beginning of Data Analysis: “The G*Power calculator [32] was used to determine the sample size. Assuming medium effect sizes (f2 = 0.15) and accounting for potential confounders (listed below), the sample size was estimated to be at least 300 participants”.

Also, the revised Data Analysis section now provides a brief explanation of model 6: “Model 6 was applied. This model allows for testing indirect effects of the independent variable on the dependent variable, assuming that the independent variable predicts the first mediator, which predicts the second mediator. In turn, the second mediator predicts the dependent variable (the outcome) [34].”

---

## [Decision Letter · Decision Letter 1]

16 Oct 2019

Links between meaning in life and physical quality of life after rehabilitation: Mediating effects of positive experiences with physical exercises and mobility

PONE-D-19-21036R1

Dear Dr. Czekierda,

We are pleased to inform you that your manuscript has been judged scientifically suitable for publication and will be formally accepted for publication once it complies with all outstanding technical requirements.

With kind regards,

Stefan Hoefer

Academic Editor

PLOS ONE

Additional Editor Comments (optional):

Reviewers' comments:

Reviewer's Responses to Questions

**Comments to the Author**

1. If the authors have adequately addressed your comments raised in a previous round of review and you feel that this manuscript is now acceptable for publication, you may indicate that here to bypass the “Comments to the Author” section, enter your conflict of interest statement in the “Confidential to Editor” section, and submit your "Accept" recommendation.

Reviewer #1: All comments have been addressed

Reviewer #2: All comments have been addressed

Reviewer #3: All comments have been addressed

2. Is the manuscript technically sound, and do the data support the conclusions?

Reviewer #1: Yes

Reviewer #2: Yes

Reviewer #3: Yes

3. Has the statistical analysis been performed appropriately and rigorously? 

Reviewer #1: Yes

Reviewer #2: Yes

Reviewer #3: Yes

4. Have the authors made all data underlying the findings in their manuscript fully available?

Reviewer #1: Yes

Reviewer #2: Yes

Reviewer #3: Yes

5. Is the manuscript presented in an intelligible fashion and written in standard English?

Reviewer #1: Yes

Reviewer #2: Yes

Reviewer #3: Yes

6. Review Comments to the Author

Reviewer #1: (No Response)

Reviewer #2: The revision was made in a comprehensive manner, addressing all issues that could be remedied and acknowledging limitations when constraints were met (design etc).

Reviewer #3: The authors have addressed each reviewer comment with care thought, resulting in an improved manuscript.

7. PLOS authors have the option to publish the peer review history of their article (what does this mean?). If published, this will include your full peer review and any attached files.

Reviewer #1: No

Reviewer #2: Yes: Efrat Neter

Reviewer #3: No

---

## [Editor Report · Acceptance letter]

23 Oct 2019

PONE-D-19-21036R1 

Links between meaning in life and physical quality of life after rehabilitation: Mediating effects of positive experiences with physical exercises and mobility 

Dear Dr. Czekierda:

I am pleased to inform you that your manuscript has been deemed suitable for publication in PLOS ONE. Congratulations! Your manuscript is now with our production department. 

With kind regards,

on behalf of

Dr. Stefan Hoefer 

Academic Editor

PLOS ONE